# Reprogramming mRNA Expression in Response to Defect in RNA Polymerase III Assembly in the Yeast *Saccharomyces cerevisiae*

**DOI:** 10.3390/ijms22147298

**Published:** 2021-07-07

**Authors:** Izabela Rudzińska, Małgorzata Cieśla, Tomasz W. Turowski, Alicja Armatowska, Ewa Leśniewska, Magdalena Boguta

**Affiliations:** 1Laboratory of tRNA Transcription, Institute of Biochemistry and Biophysics, Polish Academy of Sciences, Pawińskiego 5A, 02-106 Warsaw, Poland; irudzinska@ibb.waw.pl (I.R.); goges@ibb.waw.pl (M.C.); twturowski@gmail.com (T.W.T.); a.armatowska@ibb.waw.pl (A.A.); EWA_MAK@interia.pl (E.L.); 2Wellcome Centre for Cell Biology, The University of Edinburgh, Edinburgh EH8 9YL, UK

**Keywords:** yeast, gene expression, RNA polymerase

## Abstract

The coordinated transcription of the genome is the fundamental mechanism in molecular biology. Transcription in eukaryotes is carried out by three main RNA polymerases: Pol I, II, and III. One basic problem is how a decrease in tRNA levels, by downregulating Pol III efficiency, influences the expression pattern of protein-coding genes. The purpose of this study was to determine the mRNA levels in the yeast mutant *rpc128-1007* and its overdose suppressors, *RBS1* and *PRT1.* The *rpc128-1007* mutant prevents assembly of the Pol III complex and functionally mimics similar mutations in human Pol III, which cause hypomyelinating leukodystrophies. We applied RNAseq followed by the hierarchical clustering of our complete RNA-seq transcriptome and functional analysis of genes from the clusters. mRNA upregulation in *rpc128-1007* cells was generally stronger than downregulation. The observed induction of mRNA expression was mostly indirect and resulted from the derepression of general transcription factor Gcn4, differently modulated by suppressor genes. *rpc128-1007* mutation, regardless of the presence of suppressors, also resulted in a weak increase in the expression of ribosome biogenesis genes. mRNA genes that were downregulated by the reduction of Pol III assembly comprise the proteasome complex. In summary, our results provide the regulatory links affected by Pol III assembly that contribute differently to cellular fitness.

## 1. Introduction

Transcription in a eukaryotic cell is carried out by three RNA polymerases: Pol I, II, and III. The set of transcripts synthesized by Pol II is extremely complex because it includes thousands of different protein-coding mRNAs. In contrast, Pol I and Pol III are more specialized. Pol I is responsible for the synthesis of rRNA, and Pol III is responsible for the synthesis of tRNA and 5S rRNA, which are fundamental for the process of translation. One basic problem is how the decrease in tRNA levels caused by downregulating Pol III influences mRNA synthesis by Pol II. To address this, we used yeast mutant *rpc128-1007* located in the Rpc128 catalytic subunit near the contact points for the association between Rpc128 and the Rpc40-Rpc19 heterodimer [1]. Since the formation of Rpc128-Rpc40-Rpc19-Rpb12-Rpb10 subcomplex is the critical step in Pol III biogenesis [2,3], the *rpc128-1007* mutation has severe consequences for the assembly of the active Pol III complex and hence the ability of the cell to support Pol III transcription activity [1]. Recently published structures of human Pol III revealed a disease-relevant cluster, which is functionally similar to the yeast mutant *rpc128-1007* [4,5,6]. The mutations lead to neurodegenerative disorders called hypomyelinating leukodystrophies, and many are predicted to affect the interface between Pol III subunits, especially affecting assembly with the assembly platform, homologous to Rpc40-Rpc19-Rpb12-Rpb10 heterotetramer. Using the *rpc128-1007* mutant and a yeast model we can therefore predict how global mRNA expression will be changed by hypomyelinating leukodystrophies.

In the yeast model, genes that can suppress the cold-sensitive phenotype of the *rpc128-1007* mutant encoded the Rpb10 subunit, which is shared by all three RNA polymerases and participates in the assembly of Pol III and Pol I [7], and the fructose bisphosphate aldolase Fba1, which is involved in the control of Pol III-directed transcription [8]. The growth phenotype of the *rpc128-1007* mutant was also suppressed by the overproduction of Rbs1 and Prt1 proteins [1].

The molecular mechanism of suppression by Rbs1 relies on the improvement of Pol III assembly in *rpc128-1007* cells. Rbs1 stimulates biogenesis of the Pol III complex both directly and indirectly. First, it binds to the Pol III complex or subcomplex and facilitates its translocation to the nucleus. Following dissociation from Pol III in the nucleus, Rbs1 is exported back to the cytoplasm by Crm1 exportin [1]. Second, Rbs1 in the cytoplasm binds to a subset of mRNA and interacts with its regulatory sequences. The interaction between Rbs1 and mRNA depends on the R3H domain in the Rbs1 sequence. The binding of Rbs1 to the 3′-untranslated region (UTR) in *RPB10* mRNA leads to an increase in the synthesis of Rpb10 protein, thereby indirectly stimulating Pol III assembly [2,9].

Prt1/eIF3b is one of five subunits of the translation initiation factor eIF3 [10]. Yeast eIF3 serves as an essential scaffold that promotes the binding of other initiation factors to the 40S ribosomal subunit, where it coordinates their actions during translation initiation [11]. Considering the basic function of Prt1 in the control of translation initiation, its role in Pol III assembly appears unlikely, thus suggesting an indirect mechanism of *rpc128-1007* suppression by Prt1.

The main aim of the work was to provide a global view of a regulatory network that involves the coordination of Pol III assembly and Pol II activity. Using RNA sequencing (RNA-seq) analysis, we explored the way in which mRNA transcription by Pol II is influenced by the *rpc128-1007* mutant and whether the observed changes can be overcome by an overdose of the suppressor genes *RBS1* and *PRT1*. The reduction in Pol III assembly resulted in the extensive reprogramming of yeast Pol II genes, which are clustered in the functional classes. Expression of selected genes, examined by RT-qPCR, was specifically affected in the *rpc128-1007* mutant and suppressor strains.

## 2. Results

### 2.1. Induction of GCN4-lacZ Reporter in the Pol III Assembly Mutant Is Differentially Affected by the Suppressors RBS1 and PRT1

The *rpc128-1007* mutation has severe consequences for the assembly of the active Pol III complex and hence for the capacity of the cell to support transcription activity and growth. Reduced viability of *rpc128-1007* cells was correlated with a declined synthesis of Pol III transcripts [1]. We reasoned that a decrease in tRNA levels in *rpc128-1007* cells might result in a rearrangement of mRNA expression. A major effect that was previously observed in other mutants with Pol III machinery defects was the activation of Gcn4 transcription [12]. This regulatory response, known as general amino acid control (GAAC), relies on the translational induction of *GCN4* mRNA [13].

Hence, we investigated whether a defect in Pol III assembly similarly results in elevated Gcn4 translation. We utilized a widely employed CEN-based reporter plasmid carrying a *GCN4* promoter and 5′UTR translation regulatory sequences. Driving the expression of a lacZ gene provided a quantitative measure for Gcn4 translation by assessing β-galactosidase activity [12].

More than a six-fold induction of *GCN4*-lacZ expression was measured in the *rpc128-1007* cells with respect to the wild type under nonstarvation conditions (Figure 1A), indicating that *GCN4* translational induction was constitutively operating in the mutant with the Pol III-assembly defect. As expected, a further increase in *GCN4*-lacZ expression was observed when the cells were starved for amino acids for 3 h. These measurements are consistent with the results of Conesa et al., 2005 [12], who observed induction of *GCN4* in the majority of Pol III mutants.

To more comprehensively investigate the connection between Pol III biogenesis and *GCN4* induction, we extended our studies on the *rpc128-1007* mutant and its overdose suppressors. Two suppressor genes, *RBS1* and *PRT1*, were considered, the overdose of which complemented the cold-sensitive phenotype of the *rpc128-1007* mutant. The *RBS1* gene encodes the RNA-binding protein, which stimulates Pol III biogenesis; whereas the product of the *PRT1* gene is a subunit of the translation initiation factor, eIF3. In general, genetic suppression by *RBS1* and *PRT1* correlated with a decrease in *GCN4* induction in the *rpc128-1007* mutant to a level close to that of the wild type (Figure 1B,C). When the *rpc128-1007* cells were transformed by the plasmid that carried the *RBS1*-R3H mutant allele, in which the conserved R57 and H61 residues that are located within the RNA-binding domain R3H were changed to alanine, neither genetic suppression nor a significant decrease in *GCN4* expression was observed (Figure 1B,C, strain 4).

We supposed that the reprogramming of *GCN4* expression in the *rpc128-1007* mutant and suppressor strains is related to the levels of tRNA, especially tRNA_i_^Met^. The *rpc128-1007* strain carried additional *SUP11* and *ade1-2* mutations, which allowed us to monitor the tRNA-dependent phenotype according to the colony color. This was possible because the presence of the *ade2-1* nonsense mutation led to pigment accumulation when the dosage of the suppressor tRNA *SUP11* (Tyr/UAA) was low. The low *SUP11* dosage and the resulting red colony color were presumed to indicate that the global low tRNA levels were a consequence of a decrease in Pol III transcription in the *rpc128-1007* cells (Figure 1C, strain 2). Complementation by *RBS1* and *PRT1* resulted in a white and light pink colony, respectively (Figure 1C, strains 3 and 5), suggesting that the suppression of *rpc128-1007* due to an overdose of *PRT1* is relatively weak.

To examine the tRNA levels, small RNA species were separated on a urea-polyacrylamide gel using equal amounts of RNA per lane and stained with ethidium bromide. The level of 5.8S rRNA was comparable in *rpc128-1007* and wild-type cells and thus served as an internal control. Although 5S rRNA is also a product of Pol III, its levels are unaffected in *rpc128-1007* cells. This is unsurprising because many other Pol III mutants lead to a decrease in tRNA synthesis but do not alter the transcription of 5S rRNA [14]. As expected, tRNA levels were low in the *rpc128-1007* mutant that carried an empty vector or plasmid with a mutated *RBS1*-R3H allele and were increased by an overdose of native *RBS1* (Figure 1D). This is consistent with the R3H domain-dependent role of Rbs1 in improving Pol III assembly and the recovery of tRNA synthesis in *rpc128-1007* cells [2]. Global tRNA levels in *rpc128-1007* cells were not significantly changed by *PRT1* overexpression. This could imply that *PRT1* does not act by restoring Pol III assembly.

The same cells were examined by northern blot to quantify the amounts of single tRNAs. Levels of three representative tRNAs: tRNA_i_^Met^, tRNA^Phe^ and tRNA^Tyr^, demonstrated an approximate two-fold decrease in the *rpc128-1007* mutant relative to the wild-type control (Figure 1E). *RBS1* overexpression elevated tRNA levels in *rpc128-1007* cells; whereas the changes due to *PRT1* overexpression determined for the three specific tRNAs were not statistically significant as compared to levels in the *rpc128-1007* mutant (Figure 1E).

*RBS1* overexpression elevated tRNA levels in *rpc128-1007* cells; whereas the changes due to *PRT1* overexpression determined for the three specific tRNAs were not statistically significant (Figure 1E). In summary, *rpc128-1007* suppression by *RBS1* was correlated with the decreased activity of *GCN4* reporter and increased tRNA levels; whereas we did not observe a full correlation for suppression by *PRT1*. This difference motivated us to analyze the transcriptional response of *rpc128-1007* and suppressor mutants on a genome-wide scale.

### 2.2. RNA-seq Analysis Reveals Distinct Gene Expression Signatures of the rpc128-1007 Mutant and Its Suppressors

To compare the effects of *rpc128-1007* and its suppressors on overall Pol II-dependent gene expression, we performed RNA-seq analysis. mRNA was isolated from the *rpc128-1007* mutant that was optionally transformed with *PRT1*, *RBS1*, and *RBS1-*R3H plasmids. The same mutant that was complemented by the native *RPC128* gene, expressed from the centromeric plasmid, was used as an isogenic wild-type control (referred to as wt). Cells were grown overnight on selective medium to prevent plasmid loss, transferred to a nonselective glucose-rich medium, grown to the exponential phase, and harvested. The results of the RNA-seq experiments, performed with three biological repetitions, were subjected to differential expression analysis using the *DEseq2* method [15,16,17]. The complete data of the RNA-seq analysis are presented in Appendix A.

We first examined the overall similarity between samples by principal component analysis (PCA) (Figure 2A) and hierarchical clustering (Appendix A) for protein-coding genes. Three independent biological replicates from the same strain were clustered together, confirming data reproducibility. The analysis revealed two separate groups of strains that were genetically close: the first included the wt and the two suppressor strains *rpc128-1007* [*PRT1*] and *rpc128-1007* [*RBS1*]; the second included *rpc128-1007* and the inactive suppressor *rpc128-1007* [*RBS1-*R3H] (Figure 2A). These results correlated with the results of genetic suppression, as determined by growth under restrictive conditions. which was observed for strains from the first group but not the second group (Figure 1C).

To categorize genes that are involved in the suppression of the *rpc128-1007* phenotype, we performed hierarchical clustering of our complete RNA-seq transcriptome (Figure 2B). All genes without a significant change (*p* ≥ 0.05) in expression were annotated as unchanged and log2 of fold change was replaced with zero (*n* = 3557; white color on the heatmap). Transcription upregulation was generally stronger, exceeding a four-fold change (two on the logarithmic scale), and involved more genes than downregulation, which was characterized by a two-fold change. Groups of up- and downregulated genes, indicated by green and red, respectively, clustered together. Generally, the *rpc128-1007* and *rpc128-1007* [*RBS1*-R3H] expression profiles were similar, which is consistent with [*RBS1*-R3H] having no effect on phenotypic suppression (Figure 1C, Appendix A) and a minimal difference between both strains, as confirmed by PCA (Figure 2A).

To elucidate differences between *rpc128-1007* and the suppressed mutants *rpc128-1007* [*RBS1*] and *rpc128-1007* [*PRT1*], differentially expressed genes were clustered by expression pattern similarity between the examined strains (Figure 2C). For clustering, we only selected genes with at least one differentially expressed sample as compared to wt (*n* = 3023), and divided them into 10 clusters using a k-mean clustering algorithm. The suppressors *rpc128-1007* [*PRT1*] and *rpc128-1007* [*RBS1*] were clearly different from the *rpc128-1007* mutant strain, but they were also different from each other. These differences are consistent with their distinct effects on tRNA expression (Figure 1D,E) and, together with the phenotypic differences (Figure 1C), indicate different mechanisms of *rpc128-1007* suppression.

We analyzed genes from all clusters using annotations in the String Database (https://string-db.org/, accessed on 31 May 2021). Two top GO term functional categories for each cluster are presented in Appendix A. To validate the most interesting findings, we used an independent approach, in which the entire gene category was selected, verified using a statistical approach, and presented as boxplots (Figure 2D, Appendix A).

Genes in clusters 1 and 2 exhibited an increase in expression in the *rpc128-1007* strain (Figure 2C), which was generally compromised in the suppressor strains. Among these genes, we identified strong enrichment for amino acid biosynthesis genes (Appendix A), which are targets of the Gcn4 transcription factor. In fact, most genes from clusters 1 and 2 were under the control of Gcn4, which was noticed using the annotation of Gcn4 response genes from the published data (*n* = 511) [18,19]. Gcn4-dependent genes were upregulated in the *rpc128-1007* mutant (Figure 2D, *p* = 3.5 × 10^−40^), and suppression by either *PRT1* or *RBS1* overdose correlated with a significant decrease in their expression (*p* = 3.5 × 10^−47^ and 1.6 × 10^−25^). For *PRT1* overdose, the expression of Gcn4-dependent genes was even lower than in the wt strain (*p* = 6 × 10^−9^). Another intriguing observation was the effect of the mutated [*RBS1*-R3H] plasmid, which was used as a negative control. The overproduction of mutated Rbs1-R3H in *rpc128-1007* cells led to an increase in the expression of genes in clusters 1 and 2, which are mostly Gcn4-dependent (Figure 2D,E).

An opposite trend was observed in cluster 8, which contained functional categories that are involved in protein degradation, including 27 of the 31 subunits of the proteasome (Figure 2C,D, Appendix A). Genes that comprise the proteasome complex were downregulated in the *rpc128-1007* mutant (Figure 2D right panel; *p* = 9.6 × 10^−10^), and the suppressors *PRT1* and *RBS1* led to its partial reversal (*p* = 5.9 × 10^−8^, *p* = 8.9 × 10^−9^).

Genes that were specifically upregulated by the overexpression of *RBS1* are represented by cluster 5 (Figure 2C). Interestingly, these genes were generally unaffected in *rpc128-1007* cells but downregulated in *rpc128-1007* [*RBS1*-R3H] cells. In this cluster, we found genes involved in various functional categories. For example, genes involved in glucose transport were significantly upregulated by Rbs1 (Appendix A). These effects may have been driven by the RNA-protective function of Rbs1, mediated by Rbs1 binding to 3′-UTR regulatory sequences in mRNAs that depend on the active R3H domain in Rbs1 [2]. However, the modest and nonsignificant enrichment of Rbs1 targets that were identified by the CRAC analysis was observed among genes in cluster 5 (data not shown).

Clusters 4 and 6 represented genes that were specifically upregulated by *PRT1* overdose; however, we were unable to identify a meaningful GO category; therefore, we focused on individual genes. Among these, we found genes that encode factors that are involved in rRNA processing and biogenesis of cytoplasmic ribosomes (Appendix A). These effects were rather small; however, some increase in the expression of ribosome biogenesis genes was observed in all strains that harbored the *rpc128-1007* mutation regardless of the presence of suppressor genes (Appendix A).

Finally, cluster 7 represented genes that likely represent specific regulation that is linked to the *PRT1*-mediated suppression of the *rpc128-1007* mutation (Figure 2C). Interestingly, we identified genes that are involved in mitochondrial translation as a major functional group among genes from cluster 7 and this group strongly overlaps with group mitochondrial gene expression (Appendix A). After validation, we found that those genes were downregulated by both *PRT1* and *RBS1* suppressors (Appendix A, *p* = 2.8 × 10^−12^ and 1.3 × 10^−19^). Changes in expression for a few example genes from selected categories are given in Figure 2E.

#### Effect of *rpc128-1007* and Suppressors on the Expression of Selected Gcn4-Dependent Genes

Mostly altered by the *rpc128-1007* mutation are the Gcn4-dependent genes. Considering greater-than-two-fold changes in expression to be significant, 124 differentially expressed genes were found in the *rpc128-1007* mutant, of which 98 are under Gcn4 control (Appendix A; examples in Figure 2E). Individual Gcn4-dependent genes were, however, induced to various extents in the *rpc128-1007* mutant and their expression was differentially affected by *PRT1* and *RBS1*. The genes induced in *rpc128-1007* cells are involved in the biosynthetic pathways of amino acids (*ARG1*, *ARG3*, *HIS4*, *HIS3*, *LYS20*, *BAT1*), cofactors and vitamins (*SNZ1*, *BNA1*, *RIB5*), mitochondrial transport (*GGC1*, *OAC1*, *YMC1*), and metabolism (*ALD5*, *MAE1*, *ADH5*). One example of a Gcn4-dependent gene that was downregulated in *rpc128-1007* cells is *IGD1*, which is involved in glycogen homeostasis.

Changes in the expression of selected Gcn4-dependent genes were confirmed by direct RT-qPCR measurements (Figure 3). As expected for these genes, induction of expression was prevented by *GCN4* deletion or tRNA_i_^Met^ overproduction. Levels of *ARG1* and *ARG3* transcripts were 6.2- and 6.1-fold higher, respectively, in the *rpc128-1007* mutant than in the control (wild-type strain). Overexpression of the *PRT1* gene reduced *ARG1* mRNA and *ARG3* mRNA levels in the *rpc128-1007* mutant by 9.8- and 11-fold, approaching levels that were two-times lower than in the control (wild-type strain). Suppression by *RBS1* was relatively weaker, with 2.5- and 3.4-fold decreases in *ARG1* mRNA and *ARG3* mRNA levels being observed. The effect of the [*RBS1-*R3H] plasmid was the opposite, with *ARG1* and *ARG3* transcript levels in the *rpc128-1007* mutant increasing 2.2- and 1.7-fold, respectively. These data corroborate the RNA-seq results (Figure 2D).

However, for certain Gcn4-dependent genes in which induction was relatively weak in the *rpc128-1007* mutant, the suppression by *RBS1* was stronger than suppression by *PRT1*. This minor class of genes is represented by *OAC1*, *MAE1*, and *ALD5* (Figure 2E). For example, the induction of *OAC1* in the *rpc128-1007* mutant, which was more than a three-fold, was decreased to the level of 0.07 by *RBS1* (i.e., a 15-fold decrease) and 0.34 by *PRT1* (i.e., a 2.9-fold decrease). Moreover, no effect of *PRT1* was observed as regards the induction of *BAT1* expression, which was more than two-fold (Figure 3). Interestingly, the [*RBS1*-R3H] plasmid that encoded Rbs1 with an inactive R3H domain had no effect on the expression of these genes in *rpc128-1007* cells (Figure 2E). Altogether, while the examined genes belong to one functional category, they might be differently affected by *PRT1* and *RBS1* suppressors.

## 3. Discussion

The interplay between the three nuclear RNA polymerase systems is presumably a key aspect of growth control. Pol III synthesizes very abundant and essential RNAs whose levels might influence the expression patterns of protein-coding genes. Remarkably, Pol II is preferentially recruited in close proximity to tRNA genes that display high Pol III occupancy [20,21,22,23,24,25]. The two main aspects of the influence of RNA Pol III-directed transcription on the expression of RNA Pol II-transcribed genes in *S. cerevisiae* were studied. First, there is a positional effect that is exerted by an extremely high rate of transcription and the occupancy of Pol III machinery on neighboring Pol II genes and vice versa. The positional effect of Pol III on mRNA expression, however, only operates to a limited extent [26]; whereas Pol II transcription robustly interferes with Pol III function at specific tRNA genes [27]. Second, there is a genome-wide effect of Pol III deficiency on the expression of Pol II genes. This was revealed by microarray analyses of mutants in different components of RNA Pol III transcription machinery and of single tRNA gene deletions mutants [12,28]. Deletions of tRNA genes from single-copy families elicited a stress response; whereas deletions of genes from multicopy families led to increased expression of genes involved in translation. Other functional groups of mRNA were differentially affected by deletions of tRNA genes from single- and multicopy families [28]. 

In the present study, we used RNA-seq to analyze the expression of Pol II genes in the Pol III assembly mutant *rpc128-1007* and this study was extended to the overdose suppressors, which overcome the cold-sensitive phenotype of *rpc128-1007*. The suppressor genes, *RBS1* and *PRT1*, account for Pol III assembly and translation initiation, respectively.

The *rpc128-1007* mutant results in severe consequences for the activity of Pol III and leads to a decrease in global tRNA levels, which involved a two-fold decrease in the amount of tRNA_i_^Met^ (Figure 1). The suppression of the Pol III assembly defect by Rbs1 overdose led to a back increase in both global tRNA and tRNA_i_^Met^ levels. Although *PRT1* overexpression clearly overcame the growth phenotype of the *rpc128-1007* mutant, the correlation with the increase in tRNA levels was less prominent (compare Figure 1C and Figure 1D,E) and, therefore, functional suppression of the assembly defect by *PRT1* could be indirect. The decrease in tRNA levels in the *rpc128-1007* mutant led to a genome-wide Gcn4 response, which was differentially affected by an overdose of *PRT1* and *RBS1*, indicating that *rpc128-1007* suppression downregulates *GCN4* but occurs through two different molecular mechanisms.

Gcn4 is a transcription factor that is important for activating amino acid biosynthetic genes in response to amino acid starvation (reviewed in [29]) and regulating diverse cellular processes, including purine biosynthesis, autophagy, organelle biosynthesis, the endoplasmic reticulum stress response, and the induction of mitochondrial transport carrier proteins [18,30,31]. Gcn4 protein levels are primarily determined by translation initiation and protein degradation rather than by transcription. In yeast, the Gcn4 response relies on the derepression of *GCN4* mRNA translation and depends on the rate of ternary complex (TC) binding to 40S ribosomes. During translation initiation, methionyl-tRNA_i_^Met^ is recruited in a TC with eIF2 and GTP to the 40S subunit. This is followed by the recruitment of mRNA to form the 48S preinitiation complex, which scans mRNA for AUG recognition. The translation of Gcn4 is regulated by four small upstream open reading frames (uORFs) in the 5′ leader region of *GCN4* mRNA. *GCN4* translation is normally repressed because ribosomes dissociate from the mRNA after translation of the fourth inhibitory uORF. However, the decrease in TC concentration makes it more likely that a scanning ribosome will scan through the fourth uORF and then efficiently initiate translation on the *GCN4* sequence. In yeast that were starved of amino acids, the formation of TC is indirectly inhibited by Gcn2 kinase, which phosphorylates eIF2 and converts it to the inactive form [29]. The concentration of TC can also be decreased by the direct depletion of initiator tRNA_i_^Met^ in mutants with defects in tRNA synthesis, maturation, or nuclear export [12,32,33,34], and in a mutant with *RPL20B* deletion [35]. These mutants also derepress *GCN4* translation under nonstarvation conditions and belong to the Gcd- (general control derepressed) category.

In the *rpc128-1007* mutant that was grown under nonstarvation conditions, we observed up to a 10-fold increase in the induction of Gcn4-dependent genes that are involved in the synthesis of amino acids (*ARG1*, *ARG3*, *HIS4*, *HIS3*, *LYS20*, *BAT1*), cofactors and vitamins (*SNZ1*, *BNA1*, *RIB5*), mitochondrial transport (*GGC1*, *OAC1*, *YMC1*), and metabolism (*ALD5*, *MAE1*, *ADH5*). Therefore, the *rpc128-1007* mutant had a Gcd- phenotype. The suppression of this Gcd- phenotype by *RBS1* overdose was consistent with the enhancement of Pol III assembly and the back increase in both global tRNA and tRNA_i_^Met^ levels in *rpc128-1007* cells (Figure 1D,E). One interesting observation was the suppression of the Gcd- phenotype by *PRT1* overexpression in the *rpc128-1007* mutant (Figure 1B, Figure 3). This effect was not apparently correlated with tRNA_i_^Met^ levels, suggesting that the mechanism of *rpc128-1007* suppression by *PRT1* is different from the simple correction of the Pol III assembly defect.

Prt1/eIF3b is one of the five subunits of the translation initiation factor eIF3 [10]. eIF3 stimulates the binding of TC and mRNA to 40S subunits [36]. eIF3 also has critical functions downstream of the 48S complex assembly and prior to 60S subunit joining, which impacts ribosomal scanning on *GCN4* mRNA and AUG selection [37]. This suggests that Prt1 overdose is sufficient for the derepression of *GCN4* translation in the *rpc128-1007* mutant, with no apparent changes in tRNA_i_^Met^ abundance. Therefore, Prt1-mediated suppression was expected to result from the prevention of rescanning ribosomes to bypass uORFs when TC levels were low. Perhaps the overproduction of Prt1 altered interactions within a multifactor complex (MFC) that contained eIF3, eIF1, and eIF5 and was associated with TC [38,39]. One possible consequence of such a hypothetical rearrangement is the facilitation of TC recruitment despite its low level. Another possibility is that the Prt1 overdose affects the rearrangement of eIF3 at later stages of translation initiation. Structural studies revealed relocation of the Prt1-containing module upon mRNA binding, thus supporting a role for Prt1 in the interaction with the ribosome and selection of the AUG start codon [11,40,41,42]. Notably, the *prt1-1* mutant was previously shown to affect the Gcn4 response by altering the stringency of AUG selection during ribosomal scanning on *GCN4* mRNA [37]. The impairment in *GCN4* translation in *rpc128-1007* cells by Prt1 overdose suggests the stimulation of an as yet undefined step in the initiation of general translation machinery. Therefore, our hypothesis is that Prt1 indirectly suppresses the growth phenotype of the *rpc128-1007* mutant by improving translation and growth despite low tRNA levels.

The enhancement of the Gcn4 response in *rpc128-1007* cells by the overproduction of Rbs1 with an inactive R3H domain (Figure 2D,E) is another interesting observation. Moreover, mRNA clustering indicated the enhancement of the effect of *rpc128-1007* on genes in clusters 1 and 2, indicating a dominant-negative effect of the [*RBS1*-R3H] plasmid. The dominant mode of the *RBS1*-R3H allele was not supported by simple genetic analysis of the *rbs1*Δ strain (Appendix A). The functional negative effect of the mutated Rbs1-R3H protein that was observed herein is unclear. Rbs1 is a poly(A) mRNA-binding protein. Mutational analysis indicated that the R3H domain is required for mRNA interactions [2]. Another RNA-binding domain, SUZ, was identified in the Rbs1 sequence, but cooperation between both the R3H and SUZ domains is still obscure. Therefore, one hypothesis is that the SUZ domain alone exerts a negative effect in Rbs1 that lacks a functional R3H domain. Specific effects of Rbs1 on mRNA expression could be explained by an interaction between Rbs1 and regulatory sequences in mRNA.

The decrease in tRNA levels in *rpc128-1007* cells correlated with the downregulation of genes that are involved in protein degradation (Figure 2D), including 27 of the 31 subunits of the proteasome. Although the changes in expression were less than two-fold (Figure 2E, Appendix A), they represent a global tendency toward protein rescue. The coupling of tRNA biogenesis with protein degradation was also observed previously in yeast strains harboring deletions of single tRNA genes [28]. Deletions of tRNA genes representing the multicopy tRNA families led to downregulation of proteasomal gene expression; whereas deletions of single-copy tRNA genes have the opposite effect, i.e., upregulation. Additionally, single deletions of tRNA genes from the first group induced genes involved in amino acids metabolism and ribosome biosynthesis, but expression of these genes is decreased by deletions of tRNA genes from the second group [28]. We assume the equal transcription inhibition of all tRNA genes due to the Pol III assembly defect in *rpc128-1007* cells. However, the genome-wide effects of *rpc128-1007* mutation on mRNA expression look like those observed for deletions of tRNA genes from multicopy gene families but not for deletions of genes from single-copy gene families. 

Finally, the examination of mRNA expression for individual genes using RT-qPCR (Figure 3) generally corroborated the RNAseq results; however, it revealed diverse gene-specific effects. These may reflect the chromosomal locations of genes and the specific regulation of mRNA expression by Prt1 and Rbs1.

## 4. Materials and Methods

### 4.1. Strains, Plasmids, and Media

The *rpc128-1007* mutant MJ15-9C (MAT*a SUP11 ade2-1 ura3-1 lys2-1 leu2-3*,*112 his3*) and wild-type strain MB159-4D (control) (MAT*a SUP11 ura3 leu2 ade2-1 his3 lys2-1*) were described previously [43]. MB159-4D is referred to as wild type (wt) in Figure 1 and Appendix A.

The *rpc128-1007* mutant that harbored the native *RPC128* gene on the pRS316 centromeric plasmid pLH3 (kindly supplied by P. Thuriaux) was used as an isogenic control strain in the RNA-seq experiment and is referred to as wt in Figure 2 and Figure 3, Appendix A, and Appendix A.

The *rpc128-1007Δgcn4* strain was selected from the progeny of the cross of *rpc128-1007* mutant with the BY4741*Δgcn4* strain (Euroscrarf, Germany, Oberursel).

*IMT1* plasmid containing gene encoding tRNA_i_^Met^ cloned in Yep352 plasmid was provided by Oliver Lefebvre. YEp181-*RBS1* and YEp181-*RBS1*-R3H plasmids, referred to herein as [*RBS1*] and [*RBS1*-R3H], respectively, were described previously [2]. pFL44L[*URA3* 2μ-*PRT1*] or pFL46L[*LEU2* 2μ-*PRT1*], referred to herein as [*PRT1*], contain a cloned insert from chromosome XV originally selected in a screen of *rpc128-1007* suppressors [1]. This insert contains complete *PRT1* and the 3′ part of the *PRE10* gene. *PRT1* was essential for suppression, demonstrated by deletion of the PmlI/SalI fragment of the insert. The p180 (*GCN4-lacZ URA3 CEN*) reporter plasmid was provided by L. Valasek [44].

The following media were used for growing yeast: YPD (2% glucose (POCH, Gliwice, Poland, catalogue no. 459560117), 2% peptone (Gibco, Thermo Fisher Scientific, Waltham, MA, USA, catalogue no. 211677), and 1% yeast extract (Gibco, catalogue no. 212750); and SC (2% glucose (POCH, catalogue no. 459560117) and 0.67% yeast nitrogen base without amino acids (Difco, Sparks, MD, USA, catalogue no. 0919-15). SC-ura, SC-leu, and SC-ura-leu contained 20 μg/mL of the amino acids that were required for growth, with the exception of uracil, leucine, or both, respectively.

### 4.2. Analysis of GCN4-lacZ Expression

Strains that were transformed with p180 (*GCN4-LacZ URA3 CEN*) were grown under nonstarvation conditions at a permissive temperature (30 °C) to an optical density (OD600) of 0.6 in standard rich YPD medium (yeast extract-peptone-dextrose) or under starvation conditions (growth in YPD before being moved to minimal SC medium with no amino acids for 3 h). The cells were collected and protein extracts were prepared with a glass bead procedure [12]. β-galactosidase activity in whole-cell extracts was then measured as previously described [12].

### 4.3. Northern Blot Analysis

Overnight yeast cultures were resuspended in fresh YPD media to an OD600 of 0.2 and grown to the exponential phase (OD_600_ = 0.6). The cultures were then shifted to 16 °C, incubated for 2 h, and harvested. RNA was extracted as described previously [2]. RNA (5 μg) was separated by electrophoresis on a 10% polyacrylamide, 8 M urea gel, transferred to positively charged nylon membranes, and hybridized with DIG-labeled probes as described previously [45]. Blots were developed using higher-resolution photographic films (Medical X-Ray film blue from AGFA, Morstel, Belgium). RNA signals were calculated using ImageJ software. The following DIG-labeled probes were applied: tRNA_i_^Met(CAU)^ (5′-TCGGTTTCGATCCGAGGACATCAGGGTTATGA-DIG-3), tRNA^Phe(GAA)^ (5′-GCGCTCTCCCAACTGAGCT-DIG-3), tRNA^Tyr(GUA)^ 5′-CGAGTCGAACGCCCGAT-DIG-3′, and 5.8S rRNA (5′-GCGTTGTTCATCGATGC-DIG-3′).

### 4.4. RNA Extraction and RNA-seq Procedure

The original liquid cultures were grown overnight in SC-ura or SC-leu, transferred to YPD, and grown to the log phase (OD600 = 0.6). Total RNA was extracted from yeast cells using the Bead-beat Total RNA Mini kit (A&A Biotechnology, Gdańsk, Poland, catalogue no. 031-25BB) and purified using Clean-Up RNA Concentrator (A&A Biotechnology, catalogue no. 039-25C). RNA quality control was checked with Bioanalyzer 2100 (Agilent Technologies, Santa Clara, CA, USA).

RNA sequencing and data quality filtration were performed by the Hemispherian company with the following procedure: mRNA was enriched using oligo(dT) beads. The mRNA was fragmented randomly, and then cDNA was synthesized using an mRNA template and random hexamers primer, after which a custom second-strand synthesis buffer (Illumina), dNTPs, RNase H, and DNA Pol I were added to initiate second-strand synthesis. After a series of terminal repair and ligation of sequencing adaptors, the double-stranded cDNA library was completed through size selection and PCR enrichment.

Whole-transcriptome RNA-seq was performed using an Illumina NextSeq-500 sequencer (Novogene). Raw data that comprised three biological replicates for each strain were processed by removing reads that contained adapters and low-quality reads. The reads, having passed both operations, comprised 96% of the initial data, with an average read length of 150 bases.

### 4.5. RNA-seq Data Quantification and Analysis

The retained high-quality pair-end reads for each sample were mapped to the *Saccharomyces cerevisiae* genome sequences (version *Saccharomyces_cerevisiae*_EF4_Ensembl) using TopHat2 v 2.0.11 software on the genepattern website [46,47]. Quantitative analyses were performed using R version 3.6.3 software with the Bioconductor packages Genomic Alignments version 1.20.1 and DESeq2 version 1.24.0 [15,16,17].

The genes with adjusted values of *p* < 0.05 were considered significantly differentially expressed. Fold changes (FCs) for these genes are marked in yellow in Appendix A. The principal component analysis (PCA) plot was performed using normalized reads in the DESeq2 package [16].

For the clustering analysis, we used a subset of genes with adjusted values of *p* < 0.05. Fold changes for genes that did not pass filtration were changed to 0. Hierarchical clustering was performed using the *hclust* algorithm with the Euclidean method in the R package. Next, we removed from the analysis genes that did not have significant changes in any of the comparisons analyzed. The data were divided into 10 groups using the k-means function with default parameters. For heatmap generation, we used the *pheatmap* version 1.0.12 package from R.

To validate fold change between strains, we selected genes annotated in functional GO terms according to SGD (www.yeastgenome.org). Boxes in boxplots present data between first and third quartile (50% of observations), the line marks show median, and whiskers range between 5th and 95th (90% observations) percentile. Statistical analysis was performed using Mann–Whitney test. The boxplots were generated using ggpubr version 0.3.0 (https://github.com/kassambara/ggpubr, accessed on 31 May 2021).

### 4.6. cDNA Synthesis and Reverse-Transcription Quantitative PCR (RT-qPCR)

RNA (100 ng) was used for cDNA synthesis using the QuantiTect reverse transcriptase kit (Qiagen). cDNA for each sample was generated according to the manufacturer’s instructions. The RT-qPCR reactions contained 1 μL of cDNA template, 300 nM primer pairs, and 5 μL of RT-PCR Mix SYBR (A&A Biotechnology). Quantitative PCR was performed using the Roche LightCycler 480 System with 5-min incubation at 95 °C, followed by 40 cycles at 95 °C for 30 s, 60 °C for 20 s, and 72 °C for 20 s (with a plate read after each cycle). A melting curve analysis was performed for each sample after PCR amplification to ensure that a single product with the expected melting curve characteristics was obtained. Each sample was loaded in triplicate. Each plate contained cDNA dilutions for the standard curve, a nonreverse transcriptase control, and a no template control. Polymerase chain reaction efficiencies were between 90% and 100%. Data were processed using LightCycler 480 software and then analyzed in Microsoft Excel. Data are presented in arbitrary units, calculated from a standard curve, where the highest cDNA concentration was set to 1. Values were normalized to the levels of *ACT1* mRNA encoding actin, which was used as an internal control. The primer sequences are listed in Appendix A.

## Figures and Tables

**Figure 1 ijms-22-07298-f001:**
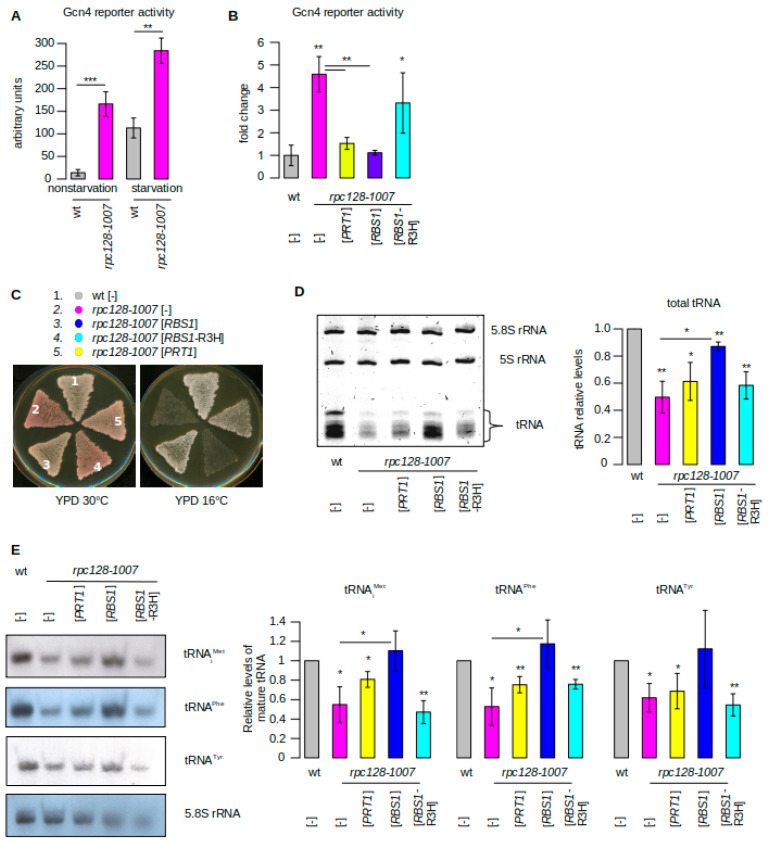
*RBS1* and *PRT1* as overdose suppressors of the Pol III assembly mutant *rpc128-1007*. (**A**) Expression of the *GCN4-lacZ* in *rpc128-1007* mutant. The *rpc128-1007* mutant and isogenic wild-type strains, transformed with p180 plasmid (*GCN4-lacZ*, *CEN*, *URA3*), were grown under nonstarvation or starvation conditions, as described in the Materials and Methods section. Extracts were prepared and assayed for β-galactosidase activity (expressed as nanomoles of *o*-nitrophenol-β-D-galactopyranoside hydrolyzed per minute per microgram of total protein). The reported values are averages of three independent measurements (number of biological replicates *n* = 3). (**B**–**D**) The *rpc128-1007* mutant was transformed with a control empty vector [–] or multicopy plasmids [*PRT1*], [*RBS1*], and [*RBS1*-R3H]. (**B**) The effect of suppressors on *GCN4-lacZ* expression. Cells harboring p180 plasmid (*GCN4-lacZ*, *CEN*, *URA3*) were grown in YPD, harvested in log phase. β-galactosidase activity was determined and calculated relative to the amounts in the wt strain, which was set as 1 (*n* = 3). (**C**) Suppression of the *rpc128-1007* growth phenotype. Cells that were grown on a YPD plate were replicated on YPD plates and incubated for three days at the respective temperatures. (**D**) Determination of total tRNA levels. Small RNA species were isolated and separated on a 7 M urea–6% polyacrylamide gel using equal amounts of RNA per lane (2.5 μg) and stained with ethidium bromide (*n* = 3). **(E)** Determination of specific tRNA levels by northern hybridization with probes specific for mature tRNA_i_^Met^, tRNA^Phe^, and tRNA^Tyr^. tRNA amounts in D and E were normalized to the loading control (5.8 S rRNA) and calculated relative to amounts in the wt strain, which was set as 1. Bars represent the mean ± standard deviation (SD) of three independent experiments (*n* = 3). Values of *p* were calculated using a two-tailed paired *t*-test: * *p* < 0.05; ** *p* < 0.01; *** *p* < 0.001. Asterisks just under bars show the *p*-value in comparison to the level in wt; other comparisons are annotated by lines. To clarify the plots, bars are colored as follows: *rpc128-1007* with control vector—pink; with [*PRT1*], [*RBS1*], and [*RBS1*-R3H]—yellow, dark blue, and light blue, respectively; grey was used for control strains. The same color code and calculation of *p*-values are used in the figures hereafter.

**Figure 2 ijms-22-07298-f002:**
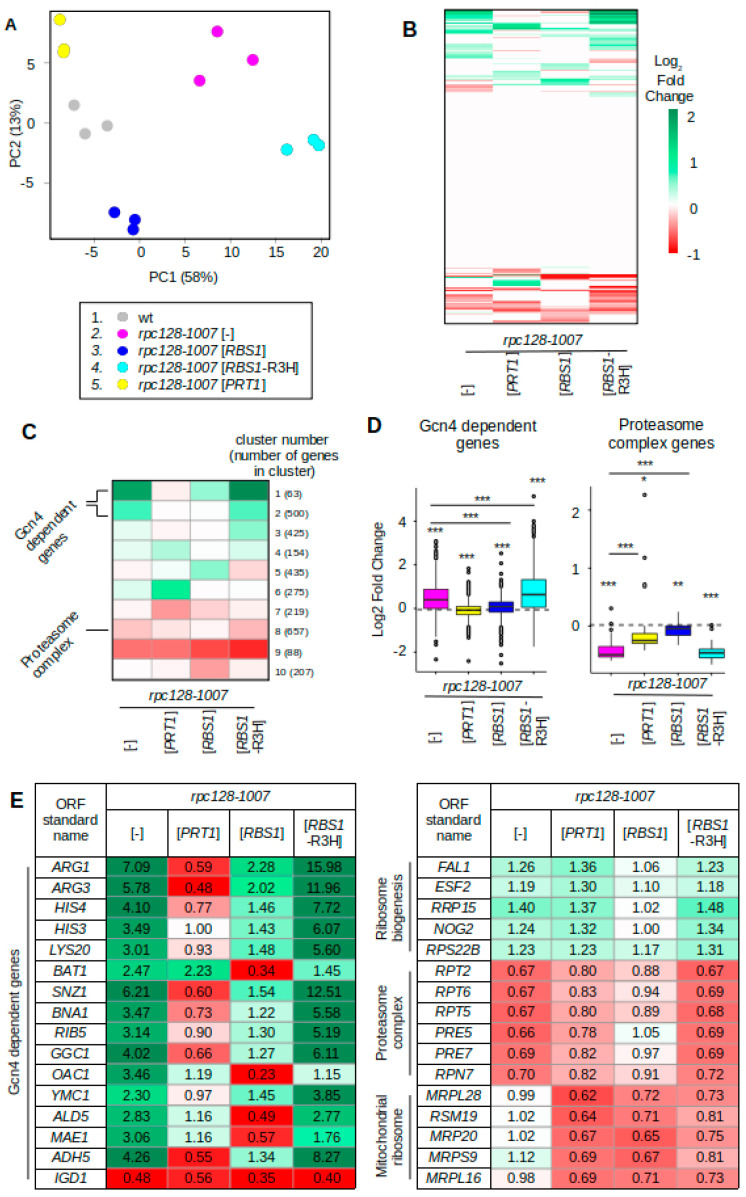
RNA-seq analysis reveals a common mechanism of *rpc128-1007* suppression by *RBS1* and *PRT1* despite functional differences. The *rpc128-1007* mutant strains were transformed with [*RPC128*], empty vector, or overexpressing [*PRT1*], [*RBS1*], or [*RBS1-*R3H] and subjected to RNA-seq analysis (*n* = 3). (**A**) Principal component analysis (PCA) of the RNA-seq data shows differences between strains and a good level of reproducibility. The axis titles show the extent of variation, which is explained by a given principal component. (**B**) Heatmap of hierarchical clustering between RNA-seq samples based on the entire yeast transcriptome. Data are shown as log2 of the ratio between each strain and wt. (**C**) Heatmap of the clustering analysis of differentially expressed genes. Selected enriched functional categories are annotated. (**D**) Boxplots of changes in mRNA levels for Gcn4-dependent genes (left) or proteasome complex genes (right). Statistical significance was calculated using the Mann–Whitney test. * *p* < 0.05, ** *p* < 0.01, *** *p* < 0.001. Asterisks just under boxes represent *p*-value in comparison to the level in wt; other comparisons are annotated by lines. (**E**) RNA-seq data for representative genes that belong to given functional categories. Data were selected from the Appendix A where the adjusted *p*-values for each expression value are specified.

**Figure 3 ijms-22-07298-f003:**
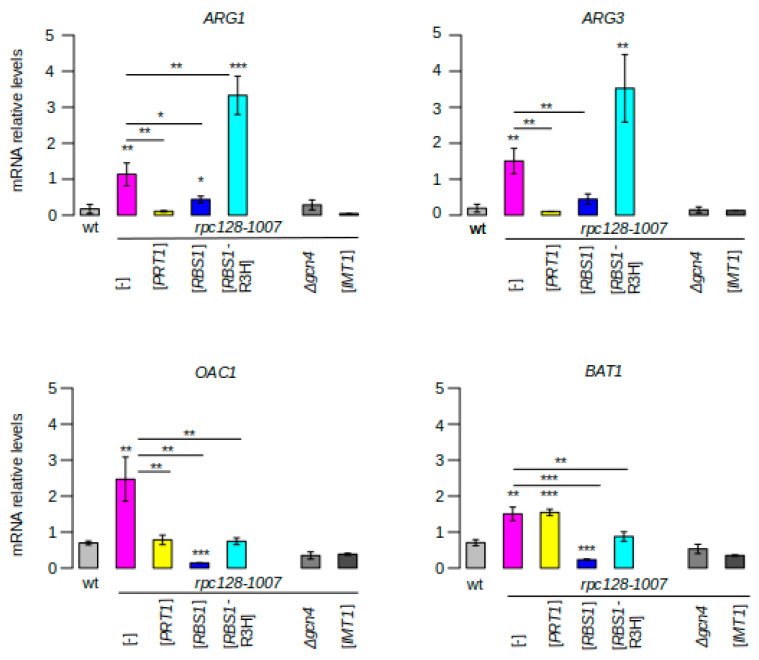
Induction of *GCN4* expression in the Pol III assembly mutant is differentially affected by the suppressors *RBS1* and *PRT1***.** RNAs that were isolated from the *rpc128-1007* mutant, which was transformed with the indicated plasmids and control strains (wt, *rpc128-1007 gcn4Δ*), were analyzed by RT-qPCR with specific probes (*n* = 3). Values of *p* were calculated using a two-tailed paired *t*-test: * *p* < 0.05; ** *p* < 0.01; *** *p* < 0.001.

## Data Availability

RNA-seq data are available on Gene Expression Omnibus (https://www.ncbi.nlm.nih.gov/geo/, accessed on 31 May 2021). Accession number: GSE166918. Data were submitted on 17 February 2021.

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
