# Peer review of "Reprogramming mRNA Expression in Response to Defect in RNA Polymerase III Assembly in the Yeast Saccharomyces cerevisiae"

_ijms, 2021, doi:10.3390/ijms22147298_

Round 1
Reviewer 1 Report
The article “Reprogramming mRNA expression in response to defect in RNA polymerase III assembly in the yeast Saccharomyces cerevisiae” by Rudzinska and colleagues describes consequences of rpc128-1007 mutation and its suppressors on yeast transcriptome. Experiments are carefully carried out. However, I have following comments:
- Please describe the number of biological replicates for each experiment. How many independent transformants was used in each case? What mean “three independent measurements” (Fig.1A and so on), technical or biological replicates?
- What reference genes were used for qPCReaction? What normalization strategy was used?
Minor comments:
- Lines 466-467, please add company names and locations
- Put reference for protein extraction protocol
- Fig.1 capitalize “e” in the legend
Author Response
Please describe the number of biological replicates for each experiment. How many independent transformants was used in each case? What mean “three independent measurements” (Fig.1A and so on), technical or biological replicates?
All results presented in the manuscript were generated in triplicate. As requested by the referee, it was stated in the figure legends: Number of biological replicates N=3
- What reference genes were used for qPCReaction? What normalization strategy was used?
Data are presented in arbitrary units, calculated from a standard curve, where the highest cDNA concentration was set to 1. Values were normalized to the levels of ACT1 mRNA encoding actin which was used as an internal control.
Minor comments:
- Lines 466-467, please add company names and locations
Information has been added as requested
- Put reference for protein extraction protocol
Reference has been added
- Fig.1 capitalize “e” in the legend
Done
Reviewer 2 Report
The paper is aimed at investigating how the defect of RNA polymerase III assembly contributes to the RNA polymerase II-dependent mRNA synthesis. In this study yeast mutant strain rpc128-1007 was characterized by applying RNA-seq analysis. The overexpression of suppressor genes rbs1 and prt1 was investigated to suppress the deleterious polymerase assembly effects. Authors observed an extensive reprogramming of yeast Pol II genes caused by the reduction of RNA polymerase III assembly.
In the present form the manuscript is very difficult to read and follow. Therefore, it is very difficult to evaluate the scientific merit of this work. The manuscript requires thorough editing of English language, particularly focusing on the style, sentence structure and logic.
Author Response
The revised version of the manuscript will be corrected by the IJMS editing service